# The Position and Complex Genomic Architecture of Plant T-DNA Insertions Revealed by 4SEE

**DOI:** 10.3390/ijms21072373

**Published:** 2020-03-30

**Authors:** Ronen Krispil, Miriam Tannenbaum, Avital Sarusi-Portuguez, Olga Loza, Olga Raskina, Ofir Hakim

**Affiliations:** 1The Mina and Everard Goodman Faculty of Life Sciences, Bar-Ilan University, Ramat-Gan 5290002, Israel; ronen.krispil@biu.ac.il (R.K.); miriam.tannenbaum@biu.ac.il (M.T.); avital1005@gmail.com (A.S.-P.); olgaloza28@gmail.com (O.L.); 2Institute of Evolution, University of Haifa, Haifa 3498838, Israel; olga@evo.haifa.ac.il

**Keywords:** circular chromosome conformation capture, genome architecture, T-DNA, transgenic, chromosomal rearrangements, synthetic biology

## Abstract

The integration of T-DNA in plant genomes is widely used for basic research and agriculture. The high heterogeneity in the number of integration events per genome, their configuration, and their impact on genome integrity highlight the critical need to detect the genomic locations of T-DNA insertions and their associated chromosomal rearrangements, and the great challenge in doing so. Here, we present 4SEE, a circular chromosome conformation capture (4C)-based method for robust, rapid, and cost-efficient detection of the entire scope of T-DNA locations. Moreover, by measuring the chromosomal architecture of the plant genome flanking the T-DNA insertions, 4SEE outlines their associated complex chromosomal aberrations. Applying 4SEE to a collection of confirmed T-DNA lines revealed previously unmapped T-DNA insertions and chromosomal rearrangements such as inversions and translocations. Uncovering such events in a feasible, robust, and cost-effective manner by 4SEE in any plant of interest has implications for accurate annotation and phenotypic characterization of T-DNA insertion mutants and transgene expression in basic science applications as well as for plant biotechnology.

## 1. Introduction

The development of efficient methods for introducing foreign DNA into plant genomes by *Agrobacterium tumefaciens* mediated transformation prompted its widespread application in biotechnology and basic science [1]. Ongoing efforts to improve the transformation technology continuously expand its use to multiple crop species and genotypes including corn, soybeans, cotton, canola, potatoes, rice, sugarcane, wheat, and tomatoes. However, the random nature of genomic integration, which is a major drawback for agricultural biotechnology, has been utilized to generate powerful resources for genetic studies by large-scale T-DNA-based mutagenesis in the model plant *Arabidopsis thaliana* [2,3,4]. The use of Agrobacterium mediated transformation is expected to increase with the developing capability of targeting T-DNA to specific genomic locations by contemporary genome editing technologies such as CRISPR/Cas9 or the use of Agrobacterium to deliver genome editing reagents to the plant cells [1,5]. 

Despite its critical significance, mapping the precise T-DNA insertion point remains a challenge due to the variable number of genomic insertion events that are frequently associated with a plethora of chromosomal rearrangements such as translocations, inversions, insertions, and deletions [6,7,8,9]. The widespread use of T-DNA-based mutagenesis and genomic engineering prompted the development of methods to map its integration sites and their associated chromosomal abnormalities. Advanced mapping technologies, from Southern blotting, cytology, and genetic mapping [10] to genomic and molecular approaches, have increased the resolution and throughput of detection capability.

PCR-based methods to detect T-DNA insertion sites, such as inverse PCR [11], thermal asymmetric interlaced (TAIL)-PCR [12], restriction PCR [13], and the biotinylated primer approach [14], use the sequence of the left and right borders of the integrated T-DNA to uncover its fusion point with the plant genome. However, given that T-DNA insertion often compromises the integrity of both plant and T-DNA sequences [9] and that the insertions include the binary vector backbone [15], the integration site may not be detected by these methods [16,17]. For example, high-resolution analysis detected small (more than 1 bp) insertions or deletions in 80% of the T-DNA borders in *Arabidopsis thaliana* [9] and small deletions of the T-DNA border were reported in ~30% of transgenic sorghum and barley plants, suggesting that the frequency of these events is high in multiple plant species [18,19]. T-DNA insertions are also associated with more complex rearrangements, such as translocations or inversions [20]. Given that T-DNA tends to integrate into several genomic loci and that in many cases only one end of the T-DNA insert is mapped, these events are more challenging to annotate. Unbiased whole-genome sequencing methods can detect both ends of the T-DNA insert in most cases despite possible sequence variations at the integration sites [21]. However, given that the size of one T-DNA insert is in the range of a few kb and that commonly the insertion consists of several T-DNA copies, uncovering chromosomal translocations by capturing the sequence continuity of the entire T-DNA insert by sequencing very long reads [15] is challenging and costly for large-scale applications. 

Chromosome conformation capture (3C)-based methods detect spatial proximity between genomic loci by measuring the relative frequency of their proximity-ligated DNA. The polymer properties of the genome dictate high contact frequency between linearly adjacent sequences. Circular 3C (4C), which profiles genomic loci in spatial proximity to a point of interest (viewpoint) [22], was applied in this study to map T-DNA insertions based on the enrichment of genomic sequences in their linear proximity. The resulting 4SEE method detects the genomic position of T-DNA inserts by capturing the local enrichment of spatial chromosomal associations in their genomic proximity without prior knowledge of their genomic locations. Moreover, deviations in this pattern were used to identify chromosomal rearrangements associated with T-DNA insertions. We analyzed several *Arabidopsis thaliana* transgenic lines and validated the location of their T-DNA insertions. Additionally, we uncovered previously unmapped T-DNA insertions and complex chromosomal rearrangements in these plants, highlighting the need for a robust and cost-efficient high-throughput method for mapping and characterizing the structure of T-DNA insertions. 4SEE is independent of the number of T-DNA copies and the genome size and thus can be readily applied to any plant with a sequenced genome. 

## 2. Results

### 2.1. 4C Signal As A Pointer for Genomic Location

Chromosome conformation capture (3C) methods [23] measure long-range chromosomal associations by enzymatic cleavage and ligation of chemically fixed nuclei. Since the resolution of 4C relies also on the frequency of sites in the genome that are recognized by the restriction enzyme (RE), chromatin was cleaved with Csp6I, a RE with a 4-bp recognition sequence (first RE in Figure 1A). In circular 3C (4C), the proximity–ligation DNA (3C) products are further processed to form DNA circles by a second digestion (DpnII; second RE in Figure 1A). This allows the selective amplification of ligation junctions with a point of interest (viewpoint) by inverse PCR (Figure 1A (. The inverse PCR primers are positioned adjacent to the ligation junction so that short (> 60 bp) reads include the viewpoint and the associating region. This allows for multiplexing of tens of experiments on a single sequencing lane. Following the mapping of sequences of the associating region to the reference genome, their local enrichment reflects long-range chromosomal association with the viewpoint. Due to the polymer nature of the chromatin fiber, the chromosomal association frequency is the highest in proximity to the viewpoint and exponentially declines as the distance increases on the linear chromosome template [24]. This feature, which is common to data derived from 3C techniques, can be applied for analysis of the linear DNA scaffold [25,26,27,28]. 

To establish both experimental and bioinformatics approaches for mapping genomic loci, we generated 4C libraries using endogenous genes as viewpoints: *ACTIN DEPOLYMERIZING FACTOR 8* (*ADF8*; At4G00680), *SBT4.12* (At5G59090), *PIF4* (At2G43010), and *MBP1* (At1G52040). Following alignment of the sequenced proximity–ligation DNA to the *Arabidopsis thaliana* TAIR10 genome, enrichment analysis was performed by assigning to the genomic RE recognition sequences a p-score reflecting the relative enrichment of signal in the surrounding 50 kb (see Methods). As expected, the highest density of RE sites with reads was at the viewpoint proximal region, allowing a simple annotation of the viewpoint position (Figure 2A, Appendix A). 

Aiming to apply 4SEE to annotate the genomic location of a viewpoint based on the position of its 4C peak, we assessed how the position of the peak center of the *ADF8* promoter is aligned with its genomic position in a wide range of 4C DNA template amounts and sequencing depths. The ligation junctions were amplified by PCR from 4C templates ranging from 30 ng to 270 ng. To vary the sequencing depth, reads containing the viewpoint sequence were subsampled from each experiment. The resulting 4C profiles demarcate a robust peak at the *ADF8* genomic position in various input amounts (Figure 2A). To assess the capability of 4C to retrieve the genomic location of the viewpoint in high resolution, we calculated the genomic coordinates of the peak base at 90%, 80%, or 50% of the maximal p-score signal. Then, for each cutoff, the distance between the center of the peak base and the genomic position of the viewpoint was calculated. A smaller distance from peak center (DPC) value indicates a smaller deviation from the viewpoint position (Figure 1B). The peak center was overall closer to the viewpoint (smaller DPC) when calculated from the base at half of the peak’s height (50% cutoff) relative to the 90% or 80% cutoff, independent of the template amounts we tested. In the 50% cutoff, the DPC maintained a robust range from ~200 bp to 4,000 bp (Figure 2B, Appendix A). As noted for the various template amounts, the center of the peak was closer to the viewpoint when calculated from the 50% cutoff relative to the 80% or 90% cutoff for the entire range of sequencing depths we tested (Figure 2C, Appendix A). In addition, the range of DPC values was much narrower in the 50% cutoff (166.5 bp to 2,407 bp, median 915.5 bp) relative to the 90% or 80% cutoff (385–11,511 bp, median 3,949.5 bp, and 32.5–12,114 bp, median 2,011 bp, respectively) for all sequencing depths and template amounts. Notably, the DPC calculated for *ADF8* in the 50% cutoff was lower than 2 kb for a wide range of tested template amounts and sequencing depths and lower than 1 kb in sequencing depths of 250 k reads or less in 60 out of 70 experiments (Figure 2, Appendix A). 

Overall, these results suggest that 4SEE is a robust tool for annotating the genomic position of a point of interest at high resolution and cost-efficiently using a wide range of starting templates.

### 2.2. Using 4SEE As A Pointer for T-DNA Insertion Sites

Having demonstrated the use of 4SEE to determine the genomic location of a locus of choice in hundreds of base pair resolutions, we sought to test its capability to map the genomic locations of T-DNA insertions in transgenic *Arabidopsis* plants. We focused on plants with single T-DNA insertions that were mapped in a base pair resolution by PCR [2]. 4C libraries were prepared from SALK_011436, SALK_009469, SALK_041474, SALK_064627, SALK_005512, and SALK_063720 homozygote plants. To capture insertions in which the left or right border of the T-DNA may have been trimmed, the 4C viewpoint we used was located in the middle of the T-DNA, ~4 kb from each border. Consequently, a large proportion of the proximity–ligation 4C signal was expected to be derived from the sequence of T-DNA insert flanking the viewpoint, thus would not provide useful positional information. Therefore, we examined whether the amount of template used to map the endogenous genes had sufficient complexity to capture the genomic fragments flanking the T-DNA insertion. To this end, we analyzed a 4C template ranging from 30 ng to 210 ng at a sequencing depth lower than 500,000 reads (Appendix A). These amounts of DNA were sufficient to capture ligation junctions at the genome flanking the T-DNA insert. The DPC calculated for these experiments did not show a clear trend related to the amount of input DNA (Figure 3, Appendix A). A median DPC of 1,132 bp calculated for the different plant lines and conditions all together reflects the accuracy of 4SEE in determining the integration position. The mean DPC for these experiments was ~3 kb, SD = 3.8 kb, reflecting the effect of outliers. These outliers do not seem to be related to a particular condition, such as the amount of input DNA, and may be inevitable given the genomic distribution of recognition sequences of the endonuclease used in 4C. 

While all the T-DNA insertions were uncovered by 4SEE at the expected positions, a robust 4C peak was detected in SALK_064627 on chromosome 1 in addition to the annotated insertion on chromosome 3, likely representing an independent T-DNA integration event that escaped previous analysis (Figure 4D,E). Overall, 4SEE demonstrates the capability to annotate the positions of all six known T-DNA insertions in these lines in kb resolution (Figure 3A and Figure 4A–D,F). 

### 2.3. Chromosomal Rearrangements Uncovered by 4SEE 

Interestingly, an additional peak in SALK_011436 was positioned ~400 kb from the known insertion site (Figure 4F). Notably, the two peaks show an asymmetric profile as the signal drops sharply at one side of the peak. This drop in the smoothed profile is clearly visualized as a sharp transition from dense to sparse 4C reads in the raw data, indicating discontinuity in the DNA that is presented on the genome browser. Such profiles indicate chromosomal rearrangements [15,29]. Given that these are the only viewpoint peaks across the SALK_011436 genome, strongly suggests that the insertion of T-DNA was associated with chromosomal inversion at this locus.

### 2.4. INTACT ADF8p:NTF/ACT2p:BirA Transgene Position and Chromatin Composition

Unlike T-DNA insertions used for genetic perturbation of endogenous genes, less attention is paid to mapping their exact positions when they are used for transgene expression. Therefore, as a test case, we aimed to annotate by 4SEE the T-DNA insertions in a widely used transgene bearing the INTACT system for affinity purification of nuclei (see [30,31,32,33,34,35]). This plant was generated by introducing a T-DNA expressing the nuclear targeting fusion (NTF) synthetic protein under the control of the *ACTIN DEPOLYMERIZING FACTOR 8* (*ADF8*) promoter (ADF8p:NTF) into a transgenic line expressing the *E. coli* biotin ligase BirA from the constitutive ACTIN2 (ACT2) promoter (ACT2p:BirA). By carrying the two types of T-DNA, NTF is biotinylated in *ADF8*-active cell types in the ADF8p:NTF/ACT2p:BirA line (CS68932). 

The integration of the ADF8p:NTF construct was uncovered by using the endogenous ADF8 promoter fragment as a viewpoint. In addition to the expected prominent peak at the ADF8 genomic position (Figure 2), four distinct 4SEE signals, representing T-DNA insertions, were found across the ADF8p:NTF/ACT2p:BirA genome (Figure 5). To validate these insertion sites, we took advantage of available deeply sequenced datasets of paired-end reads from ATAC-seq (GSE101482, 69,166,811 pairs) and DHS-seq (GSE53324, 21,138,263 pairs). We searched for reads in which one end was mapped to the T-DNA sequence and the other to the *Arabidopsis* genome as an indication of the insertion position. Strikingly, this approach revealed mate mapping to all four loci shown in Figure 5 (Appendix A). An additional T-DNA insertion site (Chr4:18,149,544) was also uncovered in the Gl2p:NTF/ACT2p:BirA line that was generated from the same parental line carrying the ACT2p:BirA insert [32], suggesting that it demarcates the ACT2p:BirA insertion. The four ADF8 insertion loci were also retrieved, but only from the ATAC-seq dataset (GSE101482), by TDNAScan software, which maps paired-end reads to the T-DNA sequence and the genome (Appendix A) [36]. 

Interestingly, in two of these loci, the 4C signal drops sharply at one side of the 4C peak (Figure 5D–E), suggesting an abnormality in the chromosome integrity in those loci potentially due to chromosomal rearrangements. Nevertheless, symmetric 4SEE peaks could reflect chromosomal abnormalities if the two ends of the rearranged loci contain the T-DNA sequence. To further understand the chromosomal structure of these loci by 4C, we analyzed genomic loci distant from the integration sites. *C2H2C* (At5G27880, Chr5: 9,885,872) is located ~53 kb from the expected chromosomal break point, as predicted by the *ADF8* 4SEE (labeled by dashed red line, Figure 6A, dark blue track). The 4C signal of *C2H2C* on chromosome 5 drops sharply at the rearrangement point in the transgenic plants, while it continues homogeneously along the chromosome in the wild-type (WT) plants (Figure 6A, light blue tracks). In addition, a high *cis*-like signal emerges in the transgenic plants on chromosome 1 at the point of a T-DNA insertion (Figure 6A, light blue tracks). The *COX11* (At1G02410, Chr1: 480,859) location is more distant relative to *C2H2C* from the T-DNA insertion (~300 kb), thus the chromosomal breakpoint is expected to deviate from the main 4C peak. Nevertheless, based on the principle that sequences located on the same chromosome in *cis* interact more frequently than sequences in *trans*, in agreement with *C2H2C* analysis, the *COX11* 4C signal drops sharply at the translocation point on chromosome 1 and continues on chromosome 5 only in the transgenic line (Figure 6A, purple tracks). Reconstruction of the fusion between chromosomes 1 and 5 at the breakpoints that were retrieved from this analysis shows a continuous and complementary 4C signal, further supporting the annotation of chromosomal translocation at this point (Figure 6B). Further support for this chromosomal rearrangement is provided by DNA fluorescence in situ hybridization (FISH) using 10-O-11 (red) and F1H21 (green) probes, positioned approximately 300 kb and 200 kb from the breakpoint on chromosomes 1 and 5, respectively. While the relative positions of the two loci are random overall in nuclei from WT plants, they constantly cluster together in nuclei from ADF8p:NTF/ACT2p:BirA plants (100 nuclei were analyzed for each plant type), indicating that they are physically linked (Figure 6C, Appendix A).

To complete the analysis of the chromosomal structure of the ADF8p:NTF/ACT2p:BirA line, we analyzed six additional viewpoints flanking the remaining three putative break sites. This revealed a complex exchange of three sub-telomeric regions between chromosomes 1 and 5 (Figure 6D, Appendix A). Notably, according to the 4C analysis, the fusions 5.2:5.4 and 5.1:5.3 did not contain T-DNA. In the absence of T-DNA sequence, the chromosomal fusion is expected to be retrieved from the dataset of paired sequences from this plant (GSE101482, GSE53324). Indeed, six nonredundant paired-end reads from ADF8p:NTF/ACT2p:BirA ATAC-seq were aligned to each chimeric DNA template that was merged at the postulated fusion points based on the 4C signal, indicating the continuity of DNA in these regions. No reads were aligned to these loci from Gl2p:NTF/ACT2p:BirA ATAC-seq data from the same lab and in similar sequencing depth, serving as a negative control (Appendix A). Overall, the ADF8p:NTF/ACT2p:BirA plant carries four inter- and intra-chromosomal rearrangements between chromosomes 1 and 5 but only two of the rearranged loci contain T-DNA. 

## 3. Discussion

In this study, we present 4SEE, which combines the application of 4C-seq in plants with a computational approach and guidelines to detect the genomic positions of T-DNA insertions together with their associated chromosomal rearrangements at high genomic resolution. 

Based on the polymer properties of the chromatin, in the close *cis* range, the spatial association frequency between chromosomal loci decays as a function of their linear genomic distance. This is reflected as a peak of high density of proximity ligation junctions at the viewpoint in the 4C-seq measurements. This peak at the viewpoint position is consistently higher than peaks from possible intra- and inter-chromosomal associations.

To increase the genomic resolution, 4C-seq was performed using 4 bp endonuclease. However, the center of the 4C peak did not align perfectly with the viewpoint due to biases introduced by the amplification of the ligation junctions by PCR and the non-uniform distribution of the recognition sequence of the endonuclease used for 4C. To minimize these biases, we calculated the local enrichment of restriction-enzyme loci having ligation junction sequences in a 50 kb sliding window. Depicting the local enrichment of positive ligation junctions rather than the number of reads was intended to reduce biases arising from PCR amplification and sequencing. The window size was aimed at averaging local asymmetry in the distribution of positive ligation junctions in proximity to the viewpoint. This local asymmetry of the 4C peak relative to the viewpoint is more pronounced at the top of the peak. Thus, the center of the peak base at 50% of the peak height was closer to the position of the viewpoint relative to the upper and narrower bases. 

In many cases, the T-DNA integration process affects both the plant and T-DNA sequences [9]. Often, the T-DNA border is truncated or includes repeats, insertions, or deletions [36]. These events may hamper detection of the T-DNA insert by using primers complementary to these sequences [16,17]. The capability to capture multiple proximity ligation junctions far from the viewpoint and use them for accurate positional mapping allows 4SEE to map T-DNAs in which the sequences at the left or right border are absent as well as inserts with multiple T-DNA copies. Thus, 4SEE uncovered additional T-DNA insertions in previously characterized SALK lines (Figure 4D,E).

Remarkably, the accuracy of positional mapping by 4SEE was not hampered using amounts as low as 30 ng of 4C template. Ideally, the viewpoint fragment provides one ligation junction per genome, including the T-DNA insert. Here, 30 ng of 4C template, reflecting approximately 100,000 *Arabidopsis* genomes, provided sufficient complexity to capture the genomic loci flanking the T-DNA insert. Nevertheless, increasing the library complexity by increasing the amount of DNA template strengthen the 4C signal at the T-DNA insertion locus (Appendix A). In addition to the amount of genomic DNA, the genomic coverage also depends on the efficiency of the molecular steps along the 4C procedure, such as enzymatic cleavage and ligation. To account for less optimal 4C experiments, the amount of DNA template should not be reduced if possible. Reducing the number of ligation junctions by sub-sampling the number of reads that contain the viewpoint sequence to 50,000 per experiment was sufficient to map the T-DNA insertion. Taking into account that only half of the single-end reads contain the proximity ligation junction, a range of 100,000 reads per plant should suffice for annotating the T-DNA position. 

4SEE resolution can reach a few hundred bp and diverge to a few kb in different measurements for the same T-DNA locus. Therefore, to annotate the T-DNA insertion site in a bp resolution, it is possible to sequence the fusion point after amplification with PCR primers for genomic loci around the 4SEE peak center. 4SEE positions for different amounts of input DNA were distributed up- and downstream to the T-DNA position. While overall approximately half of the peak centers fell less than 1 kb from the T-DNA, two measurements were more than 10 kb away (Appendix A). Thus, if data from multiple experiments are available, averaging their genomic positions while discarding outliers or calculating their median will likely increase the resolution of the T-DNA annotation to a few kb. 

Along with annotating T-DNA insertions, 4SEE uncovered chromosomal inversion in SALK_011436 and chromosomal translocations associated with T-DNA insertions in the ADF8p:NTF/ACT2p:BirA line. This line is widely used to isolate specific cell-types [30,31,32,33,34,35]. Structural analysis genomic loci flanking these insertions by 4C-seq validated these rearrangements and unraveled their complexity. Interestingly, only two of the four chromosomal fusions, involved with chromosomes 1 and 5, harbor T-DNA. Such heterogeneous T-DNA integration had been found recently in *Arabidopsis* and birch by high-throughput sequencing approaches [20,37] and may be more pronounced than previously noted.

Chromatin interactions have been used for genome assembly [25,26,27,28], and 4C was used to detect chromosomal translocations [29]. 4SEE does not require any species-specific reagent and can be applied to map T-DNA or transposon insertions [38,39] in any plant of interest by adjusting the amount of input DNA to the genome size of the analyzed plant. Annotating chromosomal rearrangements using viewpoints that are more distant from the fusion point by imbalanced distribution of ligation junctions may require deeper sequencing and higher complexity of the 4C template. 

In conclusion, we present how, in a single experiment, 4SEE is capable of detecting not only integration sites, but also structural rearrangements, such as inversions and translocations, from a small amount of NGS reads.

## 4. Materials and Methods 

### 4.1. Plant Materials and Growth Conditions

*Arabidopsis thaliana* Columbia ecotype (Col-0), SALK_011436; SALK_009469; SALK_041474; SALK_064627; and SALK_005512 from the SALK stock; and CS68932 from the Arabidopsis Biological Resource Center (ABRC) were used in this study. All plants were grown in a growth room under cool white LED light at ~100 LUX for 16 h/day at 22 °C. Surface-sterilized seeds were sown on agar-coated floats on 1/8 Murashige and Skoog (MS) media; and 14-day-old seedlings were harvested into liquid nitrogen

### 4.2. Circular Chromosome Conformation Capture (4C)

#### 4.2.1. Nucleus isolation and fixation

Frozen tissue was ground in liquid nitrogen using mortar and pestle, and transferred into nuclear isolation buffer 1 (10 mM HEPES, pH = 7.5, 0.4 M sucrose, 5 mM KCl, 5 mM MgCl_2_, 5 mM EDTA, supplemented with 0.2 mM spermine, 0.5 mM spermidine, and 0.6% Triton X-100) and incubated on a rotator at 4 °C for 30 min. The sample was then passed through 40 µm mesh and centrifuged (2500*g* at 4 °C). The pellet was resuspended with nuclear isolation buffer 2 (10 mM HEPES, pH = 7.5, 0.25 M sucrose, 5 mM KCl, 5 mM MgCl_2_, and 5 mM EDTA). The debris was further reduced by 2 rounds of gentle centrifugation at 300*g* at 4 °C for 15 sec and the supernatant was collected. The supernatant was further centrifuged for 10 min at 2000*g* at 4 °C and the resulting nucleus pellet was resuspended with NIB2. For crosslinking, the sample was incubated with 1% formaldehyde for 10 min at room temperature and neutralized with 125 mM glycine for 5 min at room temperature (RT).

#### 4.2.2. Enzymatic Digestion, Reverse Crosslinking, and Ligation

A total of 10^6^ crosslinked nuclei were washed twice with Csp6I RE buffer (Thermo Scientific, Waltham, MA, USA, #ER0211) and resuspended in Csp6I RE buffer. SDS was added for a final concentration of 0.3% (*v/v*). After 30 min of incubation at 65 °C, Triton X-100 was added for a final concentration of 1.8% and the sample was incubated for 1 h at 37 °C. Then, 150 units of restriction enzyme Csp6I (Thermo Scientific, Waltham, MA, USA, #ER0211) were added and the sample was incubated overnight at 37 °C. The inactivation of Csp6I was achieved by incubating the samples at 65 °C for 25 min. Ligation reaction was then done in 1.5 mL of 1X ligase buffer with 25 units of T4 ligase (Roche, Rotkreuz, Switzerland, #EL0011) and incubated overnight at 4 °C, followed by 30 min of incubation at RT. To reverse the crosslinks, the sample was supplemented with 50 µg of proteinase K (Invitrogen, Waltham, MA, USA, #25530049) and incubated overnight at 65 °C, followed by incubation with 150 µg of RNAse (Thermo Scientific, Waltham, MA, USA, #EN0551) for 45 min at 37 °C. The resulting proximity-ligated DNA was extracted by phenol–chloroform followed by ethanol precipitation and dissolved in 20 µl of TE buffer. To generate DNA circles, DNA was then digested overnight with DpnII (NEB, Ipswich, MA, USA, #R0543T) at 37 °C. 

#### 4.2.3. 4C Template Purification and Amplification

DNA was purified using an Expin CleanUp kit (GeneAll, Seoul, South Korea, #112-102) and DNA concentration was determined by NanoDrop 2000 (Thermo Scientific, Waltham, MA, USA). Ligation junctions were amplified by inverse PCR with viewpoint primers and Phusion hot-start high-fidelity Taq (Thermo Scientific, Waltham, MA, USA, #F-549) and the following thermal cycle conditions: 30 sec at 98 °C; 25 cycles of 98 °C for 10 sec, 60 °C for 30 sec and 72 °C for 2 min; 72 °C for 5min. PCR products were purified on Expin columns, and 30, 120, and twice 60 ng (4 amplified libraries in total) DNA templates from SALK and ADF8p:NTF/ACT2p:BirA lines were amplified separately with primer-specific barcodes. The primers were designed to hybridize as close as possible to the Csp6I and DpnII recognition sites, on the known bait sequence (genomic or T-DNA sequence). To achieve the maximum length of the capture molecule sequence (blue molecule in Figure 1A), it is recommended that the primers fit the edges of the bait sequence (red molecule in Figure 1A). To further vary the template amounts, the sequenced tags from these libraries were combined. Libraries for Illumina sequencing were prepared using NEBNext Ultra II system (NEB, Ipswich, MA, USA, #E7645, #E7103) and sequenced on a NextSeq platform (Illumina, San Diego, CA, USA). To multiplex several samples on the same sequencing lane, 4-letter barcodes were added to the primers used for inverse PCR. The list of primers is provided in Appendix A. The data are available under accession number GSE147542.

### 4.3. 4SEE Analysis of 4C-seq Data

Reads were sorted into different fastq files for each viewpoint according to the viewpoint and index sequences and aligned to the TAIR10 *Arabidopsis* genome using Bowtie [40] with the parameters –m 1 –trim primer length. To avoid possible quantitative PCR or sequencing bias, the data were transformed to represent unique coverage (> 1 reads per RE site was set to 1). To identify genomic loci with higher sequence capture frequency than expected, a p score (= −log_10_(*p*-value)) was calculated for each restriction enzyme (RE) site using a one-tailed binomial test with 50 kb running window centered at the RE site and the entire chromosome as background. Computational procedures are available at https://github.com/HakimLab/4SEE.

### 4.4. Mapping T-DNA Insertion Sites and Chromosomal Translocations from Chromatin Accessibility Data 

To map the T-DNA insertion sites from ATAC-seq and DHS-seq datasets (GSE101482, GSE53324), paired-end (PE) reads were mapped to the T-DNA plasmid sequence used to generate the ADF8p:NTF/ACT2p:BirA plant and to TAIR10 *Arabidopsis thaliana* genome using bowtie2 [41]. Reads in which one mate was mapped to the plasmid and the additional mate was mapped to the genome were extracted from the SAM file. TDNAScan [36] was used with default parameters. To find PE reads crossing chromosomal translocation junctions, reads were mapped by bowtie2 [41] to chimeric chromosomes that were generated by fusing both ends of the predicted chromosomal segments. The mapped reads were visualized using IGV software [42]. 

### 4.5. DNA Fluorescence In Situ Hybridization (FISH) 

For nucleus isolation and slide preparation, we followed [43,44] with modifications. Young seedlings, 9–14 days post-germination, were fixed in 3:1 (*v/v*) 100% ethanol:acetic acid and washed (3 × 5 min) in water. Then tissue was incubated in an enzyme buffer (10 mM citrate buffer at pH 4.6) and partially digested for 20 min at 37 °C in 6% pectinase plus 0.5% cellulase (NBC Biomedicals, UK) plus 5% cellulase “Onozuka” R-10 (Yakult Honsha Co., Ltd.) followed by washes in enzyme buffer (3 × 5 min) and distilled water (3 × 5 min). The leaves of 2 to 3 seedlings, in a small drop of water, were transferred onto a grease-free microscope slide, and the cells were spread with a stainless metal needle in a small drop of 60% acetic acid at room temperature and allowed to partially air-dry with slide rotation, then fixed in 3:1 (*v/v*) 100% ethanol:acetic acid, then immersed in absolute ethanol for 3–5 s and air-dried. For DNA probes, ABRC BAC clones (F1H21; Chr5:9635,755-9730,934, F10O11; Chr1: 491,600-603,132) were labelled by nick translation with Cy3 and Alexa Flour 488 (Cy3 and AF488 NT Labeling Kits, respectively, Jena Bioscience, Germany). FISH was conducted as previously described [43,45] with modifications. Specifically, cells were treated with 1 µg/mL of DNase-free RNase A in 2 × SSC (0.3 M NaCl plus 30 mM trisodium citrate) for 60 min at 37 °C followed by 3 × 3 min washes with 2 × SSC at 37 °C. The preparations were then dehydrated in an ethanol series (70, 90, and 100%, 3 min each) at room temperature, washed 2 × 2 min with 2 × SSC, then allowed to air-dry. The hybridization mixture (20 µL per slide under the glass coverslip 18 mm × 18 mm) contained 10% dextran sulfate, 2 × SSC, and 700 to 800 ng each of DNA probe. DNA probes and nuclei spreads were simultaneously denatured at 93 °C for 3 min and hybridized using ThermoBrite StatSpin System (Abbott, USA). Hybridization was carried out at 63 °C for 50 min. After removal of the coverslips in 2 × SSC at 63 °C, the slides were washed for 2 × 5 min in 2 × SSC at 63 °C, once in 0.1 × SSC for 5 min at 63 °C to increase stringency, then cooled to 37 °C and washed for 2 × 5 min in 0.1 × SSC, cooled to room temperature, washed in distilled water for 1 min, allowed to air-dry for 10 min, counterstained with 2.5 µg/mL DAPI for 10 min, and mounted in VECTASHIELD antifade mounting medium (Vector Laboratories). The slides were examined on a Leica DMi8 fluorescent inverted microscope using Leica Application Suite X (LAS X) software.

## Figures and Tables

**Figure 1 ijms-21-02373-f001:**
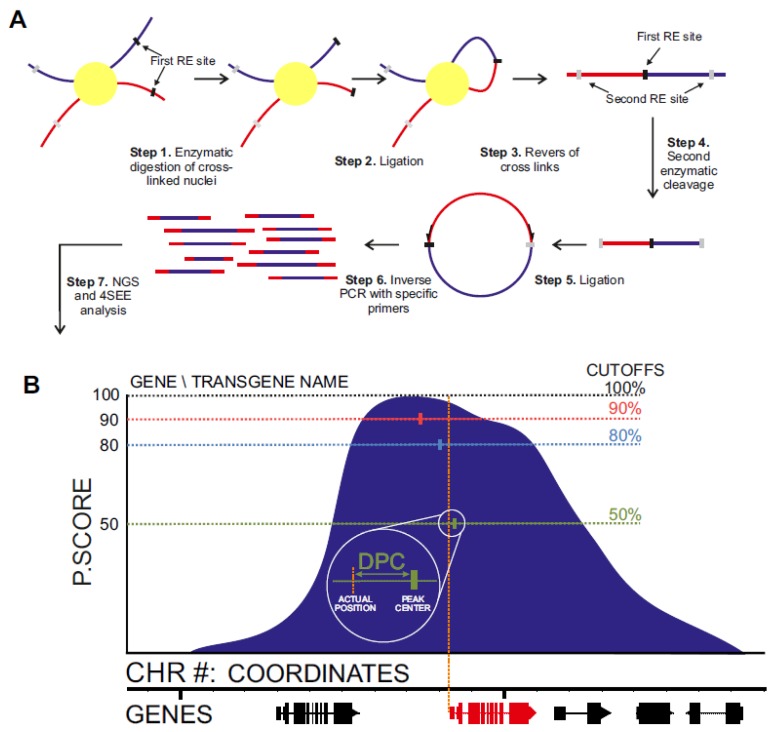
Principles of 4SEE procedure. (**A**) In circular chromosome conformation capture (4C), enzymatic cleavage and ligation of crosslinked chromatin give rise to chimeric DNA molecules between spatially proximal loci. Following reversal of the crosslinks, the DNA is cleaved with a second restriction enzyme (RE) followed by a second ligation, giving rise to circular DNA. The loci (blue) associated with the viewpoint (red) are amplified by inverse PCR and sequenced. (**B**) In 4SEE analysis, enrichment of proximity ligation products is reflected in a peak at the viewpoint position (marked by vertical red line). In this study, the distance between the viewpoint position and the center of the peak base at various cutoffs relative to the maximal p-score value (100%) of a given peak, called distance from peak center (DPC), was calculated.

**Figure 2 ijms-21-02373-f002:**
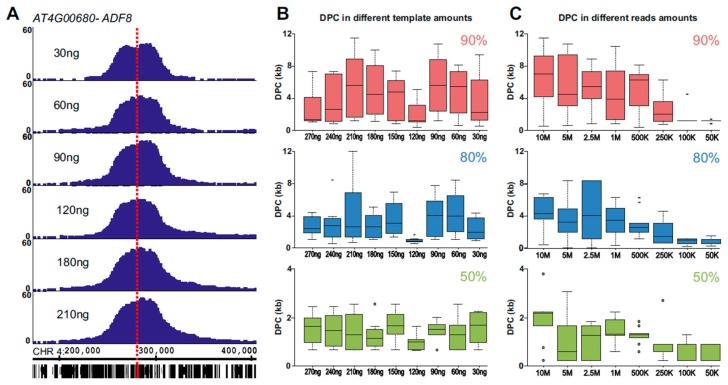
4SEE points to genomic locations of viewpoints in a wide range of input DNA and sequencing depths. (**A**) 4C peaks of *ACTIN DEPOLYMERIZING FACTOR 8* (*ADF8*) viewpoint in different amounts of 4C template. Genomic coordinates (bp) and genes are at the bottom (TAIR10). *ADF8* gene viewpoint is marked with dashed red vertical line. The *y*-axis indicates p-score. (**B**) Distance from peak center (DPC) for different amounts of input DNA. Boxplots include sequencing depths for each template amount. (**C**) DPC calculated for gradients of sequencing depths in three cutoffs. Boxplots include template amounts for particular sequencing depths.

**Figure 3 ijms-21-02373-f003:**
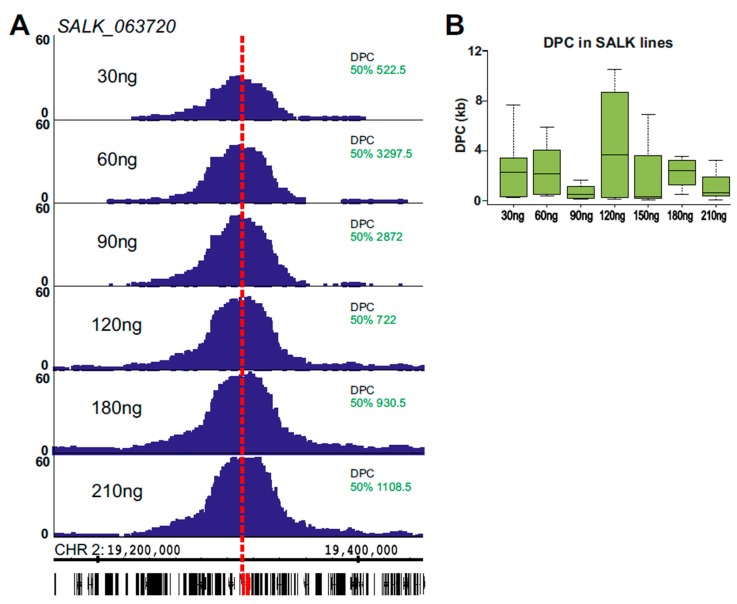
Annotation of T-DNA locations in transgenic plants by 4SEE. (**A**) 4C peaks from a gradient of 4C template amounts used for inverse PCR. Genomic coordinates and gene positions are at the bottom (TAIR10). T-DNA insertion site is marked in red, and viewpoint in vertical dashed line. DPC for the 50% cutoff is indicated. The *y*-axis indicates p-score. (**B**) DPC distribution for a gradient of template amounts in SALK lines 063720, 064627, 005512 (50% cutoff).

**Figure 4 ijms-21-02373-f004:**
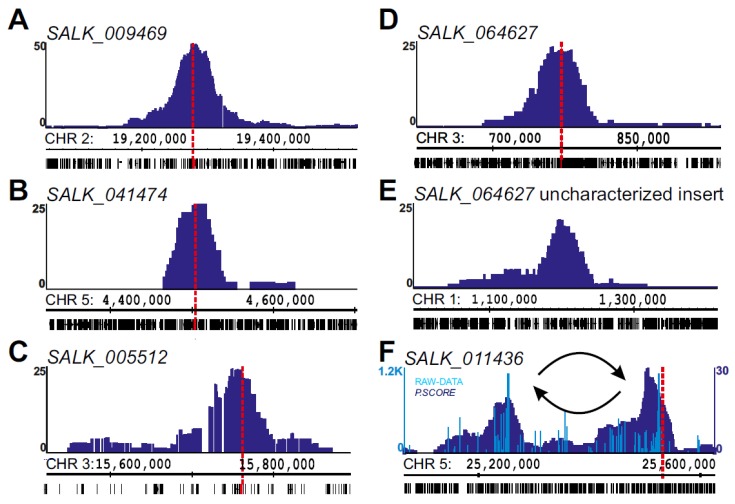
Annotating T-DNA insertions. (**A**–**D**) 4C profiles at T-DNA insertion sites (marked with vertical dashed line) in SALK lines. (**E**) Novel T-DNA insertion in SALK_064627 depicted by 4SEE. (**F**) Two 4C peaks indicate chromosomal inversion in SALK_011436. Raw data are presented in light blue bars. The *y*-axis indicates p-score. Chromosomal coordinates of *Arabidopsis thaliana* TAIR10 genome build are indicated.

**Figure 5 ijms-21-02373-f005:**
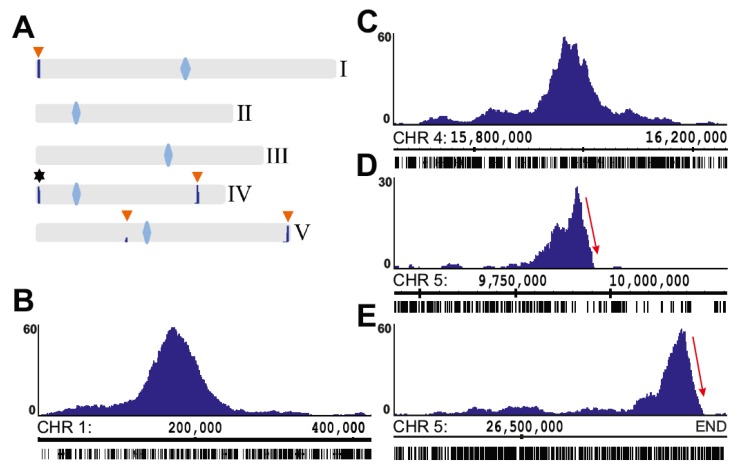
Map of endogenous and transgenic ADF8 loci. (**A**) Genome-wide map of 4SEE analysis indicating endogenous *ADF8* position (star) and four additional T-DNA insertion loci (orange arrowheads). Centromeres are marked as blue rhombuses. (**B–E**) Close-ups of T-DNA integration loci reveal asymmetric drop in signal intensity on chromosome 5 (D and E). The *y*-axis indicates p-score. Chromosomal coordinates of *Arabidopsis thaliana* TAIR10 genome build are indicated.

**Figure 6 ijms-21-02373-f006:**
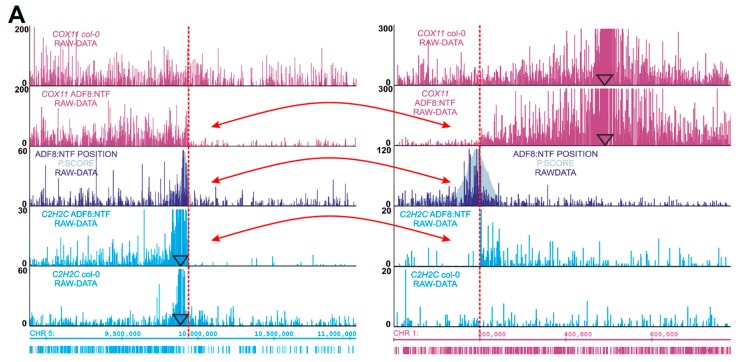
Chromosomal translocations map in INTACT ADF8p:NTF/ACT2p:BirA line revealed by 4SEE. (**A**) Translocation between left arms of chromosome 5 (fragment 5.2, left) and chromosome 1 (fragment 1.2, right). 4C signal of *COX11* on chromosome 1 continues monotonically in wild-type (WT) plants (top track), but drops sharply in transgenic plants (second track, transition from 1.2 to 1.1). This high “*cis*-like” signal appears on chromosome 5 (fragment 5.2) only in the ADF8p:NTF/ACT2p:BirA line. Transitions in 4C signal intensity occur at the T-DNA insertion position, demarcated by ADF8p:NTF/ACT2p:BirA 4C (third, dark blue track.). Similar transitions in signal intensity in transgenic plants occur for the *C2H2* viewpoint located on chromosome 5 (fourth, light blue track, fragment 5.2). The red arrows mark fusion points between the two chromosomes. (**B**) Continuous 4C profile of the three viewpoints on the reconstructed hybrid chromosome (fragments 5.2 and 1.2). (**C**) DNA fluorescence in situ hybridization (FISH) of somatic nuclei from *Arabidopsis thaliana* WT (left) and ADF8p:NTF/ACT2p:BirA transgenic plants (right). F1H21 probe on chromosome 5 labeled in green, 10-O-11 probe on chromosome 1 labeled in red, DNA counterstaining with DAPI in blue. Chromosomes at anaphase and prometaphase stages are shown for WT and ADF8p:NTF/ACT2p:BirA plants, respectively (FISH foci are indicated with arrows). Enlarged translocated chromosome is shown in the small box (**D**) Schematic diagram (in scale) summarizing chromosomal rearrangements in INTACT ADF8p:NTF/ACT2p:BirA line. Genes used as viewpoints to characterize chromosomal breakpoints and fusions are indicated alongside the chromosomes (their 4C profiles are shown in Appendix A). T-DNA inserts are noted in black. Locations of probes for DNA FISH are indicated in green and red.

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
