# Peer review of "The Position and Complex Genomic Architecture of Plant T-DNA Insertions Revealed by 4SEE"

_ijms, 2020, doi:10.3390/ijms21072373_

Round 1
Reviewer 1 Report
In this study, Krispil et al develop a method to detect T-DNA insertion sites in plant genome based on Circular Chromosome Conformation Capture (4C) and high-throughput sequencing. The authors used this method to capture T-DNA insertion sites in SALK line and transgene position. The method need a viewpoint, which is introduced by specific PCR primer to selectively amplify the ligation junctions with a point of interest. However, the method have 4 flaws: 1) the procedure of 4C method is complicate and time-consuming, including nuclei isolation, cross-linking, enzymatic digestions, reverse-crosslinking, ligations and library construction; 2) the NGS sequencing is not cost-effective for detecting several SALK lines; 3) the bioinformatics analysis is customized, which need to be open-source; 4) the method detect T-DNA insertion in kb resolution instead of bp resolution. In general, though it is not efficient, the method is somehow solid, I would encourage the method to be published if the authors could address the concerns and suggestions bellow.
In primer list of sub Table 4, the gene/locus in last row is missed. 5 primer CSP6I sites contain GTAC and 7 DPNII sites contain GATC, please explain why the other primers do not contain corresponding RE sites. Is is necessary or not? What is the principle for the primer design for viewpoint? Also, the primers for PIF4 and MBP1 endogenous viewpoint were not listed in the sub table 4. More importantly, what are the primer sequences for the 4C viewpoints located in the middle of the T-DNA? More detail of the 4SEE analysis need to be presented. In the samples of T-DNA lines, the reads coverage along the chromosome should be showed. How to distinguish the T-DNA insertion sites in local regions (kb resolution) and long term interaction regions? Further, Is it possible that the T-DNA region interact with genomic region in other chromosome? In this case, how to avoid false positive? The resolution of Figures especially Figure 1A need to be improved. The raw data need to be uploaded to GEO with accession number listed in method.
Reviewer 2 Report
All the figures in the main text appear very low in resolution. Consequently, some of the data in these figures could not properly evaluated.
More background information is needed for the circular chromosome conformation capture (4C) technique is needed in the Introduction. Particularly, what specific features of the 4C method enable achieving “detect in high resolution T-DNA insertions and the chromosomal structure of their associated genome abnormalities.” This would really help audience unfamiliar with this technique to quickly grasp the main point of this work.
Line 74, “genic location” should be “genomic location”?
Line 79, please elaborate on the “… are further processed to form DNA circles” part. I think it involves using DpnII and ligation according to the Material and Methods section. Please precisely describe the steps in the workflow for the audience to better understand this method.
Line 81, in the case of using this method to identify unknown insertion sites of T-DNA, how can the ligation junctions be determined to put inverse PCR primers? Also, the inverse PCR will require cleavage of the region between the two primers by another restriction enzyme. What are the criteria to select for an appropriate restriction enzyme for this step?
Figure 1A, could you label the “viewpoint” on this schematic picture?
Line 148 to 152, please provide the sequences of the primers used for creating 4C libraries of these SALK T-DNA lines.
Line 182 to 185, the discovery of an unannotated T-DNA insertion in SALK_064627 does not fit the title for this section, which is about chromosomal rearrangement. Suggest moving it the prior section.
Line 190 to 191, it is possible the two T-DNA fragments were inserted in tandem in this chromosomal region, rather than due to chromosomal inversion, which resulted in this signal pattern. It is unclear to me how the author had ruled out this possibility.
Figure 6C, this image is too blurred to evaluate the FISH experiment results.
Figure 6D, a white rectangle shape appears blocking some labeling underneath, please remove.
Line 345, “C” should be capital in “Col-0”
Line 346, please spell out ABRC.
Line 365, “resuspended in 1.2X Csp6I RE 1.2X.” What does “1.2X” mean and why does it appear twice?
Line 388, please make sure the sequencing data deposition is made public upon publication of this study, and the accession number is updated in the text.
Round 2
Reviewer 1 Report
In the revised version line 409-410 page 12, it still show the Data avalible (typo) under accession number GSEXXXXX. Please provide the precise GEO number instead of this. Please double check the writing to avoid typo.
Reviewer 2 Report
The manuscript has been much improved in terms of clarity. While most of the issues in the original version has been resolved, the figures in the main text are still low in resolution.
Line 111, “Csp6I, a 4 bp RE” should be “Csp6I, a RE with a 4-bp recognition sequence”
Line 173 to 176, please replace “bps” with “bp”, and keep consistent whether or not to use comma after every third digit in numbers with more than three digits.
